# Consequences of Autophagy Deletion on the Age-Related Changes in the Epidermal Lipidome of Mice

**DOI:** 10.3390/ijms231911110

**Published:** 2022-09-21

**Authors:** Yiwen Yang, Christopher Kremslehner, Sophia Derdak, Christina Bauer, Sarah Jelleschitz, Ionela-Mariana Nagelreiter, Heidemarie Rossiter, Marie Sophie Narzt, Florian Gruber, Michaela Sochorová

**Affiliations:** 1Department of Dermatology, Medical University of Vienna, Spitalgasse 23, 1090 Vienna, Austria; 2Department of Dermatology, Huashan Hospital, Fudan University, 12 Wulumuqi Zhong Road, Shanghai 200040, China; 3Core Facility Genomics, Medical University of Vienna, Spitalgasse 23, 1090 Vienna, Austria; 4Ludwig Boltzmann Institute for Experimental and Clinical Traumatology, AUVA Research Center, Donaueschingenstraße 13, 1200 Vienna, Austria

**Keywords:** epidermis, autophagy, ageing, lipidome, transcriptome, triglyceride, cholesterol ester, sphingomyelin

## Abstract

Autophagy is a controlled mechanism of intracellular self-digestion with functions in metabolic adaptation to stress, in development, in proteostasis and in maintaining cellular homeostasis in ageing. Deletion of autophagy in epidermal keratinocytes does not prevent the formation of a functional epidermis and the permeability barrier but causes increased susceptibility to damage stress and metabolic alterations and accelerated ageing phenotypes. We here investigated how epidermal autophagy deficiency using Keratin 14 driven Atg7 deletion would affect the lipid composition of the epidermis of young and old mice. Using mass spectrometric lipidomics we found a reduction of age-related accumulation of storage lipids in the epidermis of autophagy-deficient mice, and specific changes in chain length and saturation of fatty acids in several lipid classes. Transcriptomics and immunostaining suggest that these changes are accompanied by changes in expression and localisation of lipid and fatty acid transporter proteins, most notably fatty acid binding protein 5 (FABP5) in autophagy knockouts. Thus, maintaining autophagic activity at an advanced age may be necessary to maintain epidermal lipid homeostasis in mammals.

## 1. Introduction

Autophagy is a mechanism utilized by cells for recycling damaged macromolecules and for providing metabolites in situations of metabolic stress or nutrient supply shortage. This controlled process of self-digestion requires the delivery of cytoplasmic or nuclear material—the cargo—to the lysosome [1,2]. Macroautophagy, the form of autophagy investigated in this study, depends on autophagy-related (*Atg*) genes which code for machinery that can sequester material into double-membraned autophagosomes which subsequently fuse with the lysosomes where the cargo is broken down. The resulting products are used either for energy production or anabolism. Defects in or lack of autophagic capacity have emerged as, in many instances causes, or at least have correlated with many diseases, especially in cases where degradation of damaged cell components or flexible energy utilization are relevant [3,4]. Accelerated age-related decline of tissues and exacerbated age-related diseases are among the phenotypes associated with restricted or defective autophagy that are of most interest for pharmacological interventions targeting autophagy [5].

In the skin, it is well documented that autophagy decreases the rate of ageing and that functional autophagic activity declines with age [6]. In the epidermis, keratinocytes (KC) are the predominant cell type (95% of all cells) [7]. Autophagy is active both in the proliferating and differentiating KC of the human and murine epidermis [8,9]. Whereas autophagy is active during epidermal development and in the epidermal granular strata, where major remodelling of organelles and bulk degradation occurs [10], we have not found evidence for impaired removal of nuclei and other organelles in the terminally differentiating KC of epidermal autophagy-deficient (Atg5 or Atg7) mice. While Atg7 knockouts developed a thicker stratum corneum and showed sebaceous gland hyperplasia, the formation of a functional skin barrier was not impaired in epidermal autophagy knockouts [9,11].

It was recognized by Singh et al. [12] that autophagy has a role in the regulation of intracellular storage of lipids, and that deletion of autophagy resulted in increased triglyceride storage in lipid droplets in hepatocytes. The transfer of the content of lipid droplets to lysosomes and the subsequent acid lipolysis was termed lipophagy and can be utilized by adipocytes [13] and macrophages [14] among other cell types. Besides lipophagy alternative functions for autophagy in lipid metabolism have been identified, including the biogenesis of lipid droplets [15], rather than their breakdown. Thus it is necessary to determine the function of autophagy or its inhibition in the context of the organ, cell type, metabolic need, differentiation and other parameters. Our group has previously identified that a lack in epidermal macroautophagy resulting from Keratin 14 (K14) driven deletion of Atg7 did not result in a severe defect of epidermal differentiation, function or homeostasis [9]. In vitro and ex vivo studies did however suggest that under pro-ageing stress there was a profound change in lipid metabolism and an increased susceptibility to DNA damage and cellular senescence. In cultured autophagy-deficient KC, we observed increased and prolonged accumulation of oxidized phosphatidylcholines after exposure of the cells to ultraviolet A radiation [16]. We also found that the autophagy-deficient KC had strongly reduced levels of triglycerides (TAG) compared to the autophagy-competent KC. In these undifferentiated cells, the pro-senescent stress treatment with the herbicide Paraquat had also caused a decrease in TAG and an increase in free fatty acids and this effect was further enhanced in the knockouts [17].

Here we extended these studies to cohorts of young (<8 mo) and old (>16 mo) mice which did or did not lack epidermal Atg7 and studied the consequences of chronological ageing on the levels and composition of the lipidome of the epidermis. We investigated animals beyond the age of 16 months because our recent studies on atg7 deletion in sebaceous glands and tyrosinase-driven atg7 knockouts manifested the most striking and lipid-related phenotypes at an advanced age [11,18]. Here we found an age-related increase in triglyceride levels in the mouse tail epidermis that was significantly less pronounced in the autophagy-deficient old animals. Quantification of cholesterol ester classes showed a significant decrease in the esterified long-chain polyunsaturated fatty acids in the knockouts. The content of sphingomyelins (SM) with acyl chain lengths of C18-C24 decreased in the aged epidermis, and this effect was strongly enhanced in the autophagy knockouts, which additionally had significantly decreased content in SM with shorter acyl chain (C14–C17) species.

In the transcriptome of corresponding tail epidermal keratinocytes, we found the fatty acid transporter Fatty Acid Binding Protein 5 (FABP5) as the gene most highly induced in Atg7 depleted tissue.

Immunohistological analysis of FABP5 expression showed an age-related change in epidermal staining distribution and, in contrast to gene expression in the monolayer cultured keratinocytes, a massive decrease in FABP5 staining in the old Atg7 epidermal knockout animals compared to controls. Together the results demonstrate the role of autophagy in the regulation of triglyceride synthesis and of the fatty acid composition of cholesterol esters and sphingolipids of the epidermis in ageing.

## 2. Results

### 2.1. Aged Autophagy Deficient Epidermis Accumulates Less TAG than Aged Autophagy Competent Epidermis and Shows Dysregulation of Other Lipid Classes

In this study, we investigated the tail skin of mice in which the Krt14 (Keratin 14) promoter-driven Cre-LoxP system deletes a floxed exon of the essential autophagy gene, Atg7 (atg7ΔKC, knockout) in comparison to tail skin of control mice (Atg7^F/F^, controls).

We collected epidermis from tail skin of young (<8 months) and old (>16 months) mice of both genotypes, isolated total lipids and performed targeted high-performance liquid chromatography–mass spectrometry (HPLC-MS) lipidomics of the epidermal lipid extracts. We then determined the abundance of selected major and minor lipid classes, normalized to the wet weight of the epidermis prior to extraction. Among the lipid classes, we found a striking increase of TAGs in the epidermis of aged control mice as compared to the young controls. This age-related accumulation of TAG was significantly reduced in the aged atg7ΔKC, however, their levels were still significantly upregulated compared to the young groups (Figure 1A). Next, we analyzed in detail the changes within the individual lipid classes, focusing on differences between aged animals and their young counterparts (Figure 1(Bi)) and between animals of different genotypes but of comparable age (Figure 1(Bii)). We found that there was a decrease in cholesterol esters (CE, significant only for young knockouts) and in sphingomyelins (SM) for the knockouts of both age groups when compared to the age-matched controls (Figure 1(Bii)). In the aged atg7ΔKC in comparison to young atg7ΔKC, a significant reduction was observed for phosphatidylcholines (PC), ether-PC (PC-O), lyso-PC (LPC), ether-LPC (LPC-O), phosphatidylethanolamines (PE) and ether-PE (PE-O). A significant decrease in LPC-O and lyso-PE (LPE) was also observed in aged versus young controls (Figure 1(Bi)). In summary, we observed a general decrease in phospholipid classes in the old knockout but not old control mice compared to their genotype-matched controls. On the other hand, in age-matched groups, Atg7 deficiency reduced levels of glycerolipids. As not only the total amount but also the lipid acyl chain length and its saturation influence the physico-chemical behaviour and biological function of lipids, we further analysed the species within the regulated lipid classes.

### 2.2. Autophagy Deficiency Decreases TAG Unselectively but Leads to Selective Reduction of Epidermal CE-Esterified with Long Chain PUFA and to Reduced Abundance of SM Species

A closer evaluation within lipid classes showed that autophagy deficiency reduced the age-related upregulation of most of the TAG species in a similar way Figure 2A,B (for mean values ± SD see Appendix A). This could suggest that the autophagy deficiency either leads to increased TAG lipolysis or to their decreased synthesis and therefore decreased accumulation.

Within the CEs and SMs, the individual species were affected differently by age and/or genotype. Figure 2C (CEs) and Figure 2E (SMs) display an overview of the absolute amounts of these lipids, grouped by acyl chain length. Using analysis of variance (Figure 2C), the most abundant CE (CE 20:5, cholesterol esterified with eicosapentaenoic acid), was significantly decreased in both knockouts compared to corresponding controls. When comparing just two groups separately (evaluating the effect of either age or genotype) we observed that among the CE species investigated there was a significant decrease in the polyunsaturated CE 20:4, CE 20:5 and CE 22:6 in the autophagy-deficient epidermal samples (Figure 2(Dii); CE 20:5 and CE 22:6 were reduced only by the trend in the old group). Since the polyunsaturated fatty acids, which can be bound to the cholesterol moiety, are important intermediates in many biochemical processes (e.g. inflammation, synthesis of intermediates), the decrease in CE with these acyl chains could indicate an increased FA availability for other processes.

From the SM species investigated several observations can be made. Firstly, there is an apparent trend toward reduction in their absolute abundance in both knockout groups for all SMs, with a stronger effect in the aged cohorts, and for acyl chain lengths C14–C20 (significant for C15, 16, 18 and 20; Figure 2E, analysis of variance). These changes are also visible in Figure 2F (effect of either age or genotype): the species with the acyl chain length of C19–C23 were by trend reduced in the aged Atg7^F/F^ control cohort, the Atg7 deletion resulted in a further (significant) reduction of the same species and of species with C17 and C18 in the aged atg7ΔKC cohort, when compared to the corresponding young counterparts (Figure 2(Fi)). The atg7ΔKC also showed a significant age-independent decrease in the short chain SM species, C14 and C15 (Figure 2(Fii)). Furthermore, SM C16–C18 in aged and C18, 19, 21, 22 in young knockouts were significantly decreased when compared to their age-matching controls (Figure 2(Fii)). SMs are one of the precursors of selected ceramides (Cer) [19], yet the total amount of Cer was affected neither by age nor by autophagy knockout (Figure 1A). Nevertheless, Cer species with acyl chains of C18 and C22 showed a tendency to increase, and Cer with long acyl chain (C24) tended to decrease in atg7ΔKC young epidermis (Appendix A), possibly partially due to the observed changes in SM levels.

### 2.3. Autophagy Deficiency Affects Transcription of Lipid/Fatty Acid Transporters and Inflammatory Genes in Cultured Keratinocytes

We analyzed the expression of genes involved in lipid metabolism in cultured keratinocytes isolated from the epidermis of young and old atg7ΔKC and Atg7^F/F^ mice (*n* = 4 per group) to identify whether the observed changes in lipid abundance could be caused by changes in gene expression. Transcripts of lipid-related candidates were screened in the differentially expressed genes (DEGs) enriched in ‘lipid’-related terms in ‘Diseases and Biological Functions’ by Ingenuity Pathways Analysis (IPA) and ‘Biological Process’ of Gene ontology by Metascape from 4 comparisons: old Atg7^F/F^ vs. young Atg7^F/F^, old atg7ΔKC vs. young atg7ΔKC, young atg7ΔKC vs. young Atg7^F/F^ and old atg7ΔKC vs. old Atg7^F/F^, thereby identifying 55 lipid-related genes whose expression was significantly changed.

A clustering heatmap was plotted for showing the expression pattern of DEGs in different samples. While the clustering of the lipid-related transcripts separated atg7ΔKC from Atg7^F/F^ and further segregated the young Atg7^F/F^ groups from the old Atg7^F/F^ groups, the old and young atg7ΔKC were not segregated (Figure 3A).

Next, we restricted the DEG to those genes where at least one condition displayed a “Transcripts Per Kilobase Million” (TPM) value of more than 50, typically indicating a medium to high expression level [20] (Figure 3B), and those with 10 to 50 TPM in at least one condition but discarding transcripts with very low expression (Figure 3C). The most prominently induced DEG was FABP5, a lipid chaperone that was reported to be involved in the uptake, intracellular transport and metabolization of fatty acids [21]. The FABP5 expression increased markedly in autophagy-deficient groups, especially in the young autophagy-deficient group (Figure 3B)**.** Apolipoprotein E (ApoE), a transporter of lipoproteins that is also expressed in the basal epidermis [22], on the other hand, was downregulated in autophagy-deficient cells (Figure 3C). Several of the other lipid-related DEG induced in the autophagy-deficient keratinocytes were pro-inflammatory genes, several of them involved in neutrophil chemotaxis. These were the cyto- and chemokines interleukin 24 (IL24) and C-X-C motif ligand 3 (CXCL3), adipose glutathione S transferase (GSTA4), and lipocalin 2 (LCN2), as well as prostaglandin E synthase (PTGES), the products of the latter regulating neutrophil function [23,24,25]. Among them, PTGES and CXCL3 (only by trend) expression increased in autophagy-deficient groups compared with the autophagy-competent groups. On the other hand, LCN2 expression increased in old mice compared with young ones. IL24 and GSTA4 expression were affected by both age and autophagy competence and showed increases in autophagy-deficient groups and decreases in old groups (Figure 3B,C).

### 2.4. Fatty Acid Binding Protein 5 (FABP5) Expression Is Strongly Reduced in the Epidermis of Old atg7ΔKC

As we hypothesized that a massive change in Fabp5 expression and localization could yield insights into the mechanism behind the age- and autophagy-mediated changes in the epidermal lipidome, we immunostained cross sections of tail skin for Fabp5. We chose sections in the scales of the mouse tail skin that contained comparable positions relative to the pilosebaceous units and found that the sebocytes of the sebaceous gland showed strong positive Fabp5 staining in all sections. Fabp5 was expressed evenly in the suprabasal epidermis in the young animals of both genotypes, yet the strong increase in mRNA expression of Fabp5 in cultured autophagy-deficient cells was not observed in the immunofluorescence staining in the epidermis of neither young nor old autophagy-deficient animals. The Fabp5 staining intensity was more concentrated in the stratum granulosum of old Atg7^F/F^ animals whereas the epidermis of old atg7ΔKC animals was largely devoid of staining, as shown by quantification with image analysis software (Figure 4). Given the results from the transcriptomic analysis where FABP5 was strongly increased in atg7ΔKC cells (that were, however, proliferating in culture), this finding was unexpected.

## 3. Discussion

### 3.1. Autophagy in Ageing

The role of autophagy in skin ageing has been a matter of discussion, as it is a frequent observation that autophagic flux decreases with age [26]. On the other hand, it has been suggested that autophagy activation is required for inducing cellular senescence, a cell ageing phenotype also identified in the skin and more frequently present in aged skin [27], including epidermal keratinocytes [28]. The effect of autophagy on ageing strongly depends on which of the specific cell types present in the skin is affected by the decline or activation of autophagic activity [6]. Whereas stem cells require autophagy for homeostasis and its lack impairs the supply of functional progeny, the short-lived differentiating cells—in the skin these are most prominently the proliferating and differentiating epidermal keratinocytes—appear to utilize autophagy mostly as an induced stress response. On the other hand, long-lived differentiated cells, such as dermal fibroblasts, require autophagy for sustaining cell function by detoxification and energy supply maintenance during ageing [29,30]. Our findings reported here suggest that in the epidermis there may be a role for autophagy in contributing to age-related adaptations by affecting triglyceride—storage lipids—but also by inducing specific changes in the fatty acid composition of other lipid classes.

### 3.2. Autophagy and Lipid Homeostasis—Triglycerides

Not only the lysosomal, acidic lipolysis but also neutral cytosolic lipolysis can be affected by components of the autophagy machinery—for example, adipose triglyceride lipase (ATGL) has an interaction domain with the autophagosomal coating protein LC-3 and mutation of this domain disrupts the interaction of ATGL with lipid droplets. Furthermore, autophagy may not only act as the catabolic mechanism for lipid droplets but in certain settings can also be required for the formation/biogenesis of lipid droplets and subsequent fatty acid oxidation independent of lipophagy [31,32]. Finally, a role for autophagy in the phagocytosis of lipoproteins has also been suggested [33]. This may be relevant in the transport and degradation not only of native but also of oxidized lipids in autophagy, a field that we have studied earlier [16]. When investigating myocardial energy metabolism in autophagy-deficient mice Altamimi et al. recently found that these mice had a reduced triglyceride reservoir, which they suggested was due to reduced FABP5 expression and therefore reduced fatty acid uptake/translocation [34]. As we found reduced accumulation of triglycerides and a strong decrease in FABP5 protein in the aged epidermis of atg7ΔKC mice (despite the increase in the transcript in the proliferating cultured KC), these factors and the utilization of fatty acid oxidation versus glucose utilization will be interesting to investigate. ApoE binds to lipoproteins which are then taken up by tissue receptors, a process which clears lipoproteins from plasma [35], and it is expressed in rodent skin [36]. In human skin, ApoE is also expressed in the basal layer of the epidermis and in basal areas of the pilosebaceous unit, and ApoE can be secreted by keratinocytes, however, its function in epidermal or sebaceous lipid homeostasis is unknown [22]. The downregulation of a lipoprotein trafficking factor for basal epidermal cells and the upregulation of a fatty acid trafficking factor that is normally expressed in suprabasal cells together with the observed changes in the epidermal lipidome warrants further investigation. Of course, the parameters age, epidermal differentiation and autophagic activity require that such questions be answered in a spatio-temporal (regarding organismal and cellular ageing) context.

### 3.3. Autophagy and Lipid Homeostasis—Cholesterol Esters and Sphingomyelins

Our MS analytical method did not cover free fatty acids in the lipid extracts but allowed observations on FA esterified in TAG, CE and SM. Here we found no significant difference in the chain length and double bonds of the TAG esterified fatty acids, but a very strong decrease of FA 20:4, 20:5 and 22:6 esterified to CE. An autophagy-specific reduction of the short chain acyls in SM, and an age-dependent reduction in their C19- to C23-long acyls were significantly more strongly reduced in old atg7ΔKC.

Sphingolipids affect autophagic function since ceramides control cellular nutrient uptake and therefore the mTOR-dependent signalling that modulates autophagic activity. In addition, they directly affect the fusion of autophagosomes with lysosomes and Atg5 cleavage at the initiation of autophagosome biogenesis [37]. Chain-length-specific effects of sphingolipids on autophagy regulation have been reported in several tissues [38]. Ceramides from C14 to C20 are elevated upon induction of autophagy by photodamage [39], an inducer of autophagy that also selectively affects the restoration of phospholipid homeostasis after challenge and the loss of which reduces the lysophospholipid levels, as we have found in cultured atg7ΔKC [16]. An interesting finding regarding sphingolipids and an autophagic secretory pathway was made recently, showing that loading of specific extracellular vesicles requires neutral sphingomyelinase 2, an enzyme needed for epidermal barrier lipid homeostasis [40], and the LC3 conjugation machinery [41].

Regarding the CE-esterified PUFA, further studies need to clarify whether these are synthesized or utilized more, or (per)oxidized in the autophagy knockouts. Interesting in that respect is that FABP5 preferentially translocates PUFAs, specifically arachidonic acid, to the nucleus [42] for PPAR but also NfKb signalling. This, together with their uptake, and their availability for CE esterification, would be compatible with increased pro-inflammatory gene expression in the autophagy knockouts as one of many possible explanations. Fabp5 was shown to mediate cytokine expression not only in macrophages but also in keratinocytes of mice after a high-fat diet, and this effect was mediated by saturated fatty acids in the macrophages [43]. The authors discuss a connection between autophagy and the saturation of fatty acids with the activation of the inflammasome and cytokine production. From our findings, the activation of downstream genes for eicosanoid synthesis and the metabolization of the PUFA by them to inflammation mediators may, together with the expression of other lipid-related chemokines, contribute to a more inflammatory state, as has been observed in autophagy-deficient cells or tissues [3,44]. Why the FABP5 immuno-reactivity is strongly reduced in the aged tail epidermis sections of atg7ΔKC compared to the controls also remains elusive for now, but suggests that the age-dependent effect of autophagy on FABP5 protein depends on differentiation or the epidermal context. Furthermore, conformational changes upon binding or oxidation (FABP5 is highly prone to oxidation, which can destabilize the protein [45], and we have previously found more redox stress in atg7ΔKC [11]) and reduced translation or increased export are hypotheses to be followed in the future.

Whereas this study confirmed that at least in homeostasis epidermal autophagy deficiency does not cause obvious or severe defects in the epidermis in aged mice, it identified that age-related homeostasis of epidermal lipids with known biological activities requires functional autophagy and identified genes dysregulated in autophagy-deficient skin that potentially mediate the observed changes in the lipidome or the downstream signalling events.

Our findings provide a rationale for further investigation of the role and the mechanistic base of autophagy-dependent lipid synthesis, mobilization, transport and degradation in regulating inflammation, barrier function and energy metabolism of the epidermis in ageing.

## 4. Materials and Methods

### 4.1. Atg7^F/F^ and atg7ΔKC Mice

The Atg7^F/F^ mice, which served as controls, the atg7F/F Krt14-Cre (termed “atg7ΔKC”, or “mutant” test mice), in which Atg7 is inactivated, have been described previously [9]. Animals were kept under standard housing conditions, with access to food and water ad libitum, and 12-h light and dark cycles until they reached the indicated ages. They were killed by cervical dislocation according to the guidelines of the Ethics Review Committee for Animal Experimentation of the Medical University of Vienna, Austria. Genotyping from toe-derived genomic DNA for the floxed Atg7 and cre genes was performed as in [9].

### 4.2. Lipid Extraction

The samples were weighed in 0.5 mL Precellys® CK14 Lysing Kit tubes (Bertin Instruments, Montigny-le-Bretonneux, France) and methanol was added (3 µL methanol/1 mg tissue). The samples were homogenized using a Precellys 24 tissue homogenizer (Bertin Instruments, Montigny-le-Bretonneux, France) equipped with a Cryolys cooling unit. Lipids were extracted using a modified methyl-tert-butyl ether (MTBE) method. In brief, 20 µL of the sample homogenate were transferred into a glass vial, 10 µL internal standard solution ((SPLASH® Lipidomix®) from Avanti Polar Lipids (Alabaster, AL, USA)) and 120 µL methanol was added. After vortexing 500 µL MTBE was added and the mixture was incubated in a shaker for 10 min at room temperature. A phase separation was induced by adding 145 µL MS-grade water. After 10 min of incubation at room temperature, the samples were centrifuged at 1000× *g* for 10 min. An aliquot of 450 µL of the upper phase (organic) was collected and dried in a vacuum concentrator. The samples were reconstituted in 200 µL methanol and used for LC-MS analysis.

### 4.3. Lipidomics Analysis

Analysis of lipids using LC-MS was performed at the CeMM core facility: The LC-MS analysis was performed using a Vanquish UHPLC system (Thermo Fisher Scientific, Waltham, MA, USA) combined with an Orbitrap Fusion™ Lumos™ Tribrid™ mass spectrometer (Thermo Fisher Scientific). Lipid separation was carried out by reversed-phase chromatography employing an Accucore C18, 2.6 µm, 150 × 2 mm (Thermo Fisher Scientific) analytical column, column temperature was set to 35 °C. Acetonitrile/water (50/50, *v/v*) solution containing 10 mM ammonium formate and 0.1% formic acid was used as mobile phase A. Mobile phase B consisted of acetonitrile/isopropanol/water (10/88/2, *v/v/v*) containing 10 mM ammonium formate and 0.1% formic acid. The flow rate was set to 400 µL/min and a gradient of mobile phase B was applied to ensure optimal separation of the analyzed lipid species. The electrospray ionization in positive and negative mode was used for MS analysis, the MS conditions were as follows: capillary voltage, 3500 V (positive) and 3000 V (negative); vaporize temperature, 320 °C; ion transfer tube temperature, 285 °C; sheath gas, 60 arbitrary units; aux gas, 20 arbitrary units; sweep gas, 1 arbitrary unit. The Orbitrap MS scan mode at 120,000 mass resolution was employed for lipid detection. The scan range was set to 250–1200 *m*/*z* for both positive and negative ionization mode, the AGC target was set to 2.0 × 10^5^ and the intensity threshold to 5.0 × 10^3^. The data-dependent MS2 scan using HCD with fixed collision energy mode set to 35% and inclusion list was employed to obtain MS2 spectra for Cer and Cer-PE lipid species. The data analysis was performed using the TraceFinder software (Thermo Fisher Scientific).

### 4.4. Primary Keratinocyte Culture from Mouse Tail

Primary keratinocytes were isolated from the tails of young and old Atg7^F/F^ or atg7ΔKC mice as described before [16] (young: <8 Months; old: >16 Months). The cells were cultured with keratinocyte growth medium-2 (KGM-2; Lonza, Basel, Switzerland) and experiments were performed without further passaging.

### 4.5. RNA Isolation, RNA Sequencing (RNA seq) and Heat Map Generation

Total RNA was extracted from mouse keratinocytes (*n* = 4 for each group) with Buffer RLT Lysis buffer (Qiagen, Düsseldorf, Germany) according to the manufacturer’s instructions. The RNA cleanup and concentration were performed using the RNeasy Cleanup Kit (Qiagen) according to the manufacturer’s instructions. Sequencing libraries from the total RNA of the samples were prepared at the Core Facility Genomics, Medical University of Vienna, using the QuantSeq FWD protocol (Lexogen, Vienna, Austria). Optimal numbers of PCR cycles were determined by qPCR according to the library prep manual. Libraries were QC-checked on a Bioanalyzer 2100 (Agilent, Santa Clara, CA, USA) using a High Sensitivity DNA Kit for correct insert size and quantified using Qubit dsDNA HS Assay (Invitrogen, Waltham, MA, USA). Pooled libraries were sequenced on a NextSeq500 instrument (Illumina, San Diego, CA, USA) in 1 × 75 bp single-end sequencing mode. Approximately 7 million reads per sample were generated. Reads in fastq format were generated using the Illumina bcl2fastq command line tool (v2.19.1.403). Reads were trimmed and filtered using cutadapt [46] version 1.15 to trim polyA tails and remove reads with N’s. After cleanup, reads in fastq format were aligned to the mouse reference genome version GRCm38 with Gencode mV23 annotations using STAR aligner [47] version 2.6.1a in 2-pass mode. Reads per gene were counted by STAR, and differential gene expression was calculated using DESeq2 [48] version 1.22.2.

Adjusted *p*-value < 0.05 and absolute log2FoldChange (FC) of at least 1 were set as the threshold for DEGs.

Lipid-related candidates were screened in the DEGs enriched in ‘lipid’-related terms in ‘Diseases and Bio Functions’ by Ingenuity Pathways Analysis (IPA) and ‘Biological Process’ of Gene ontology by Metascape (https://metascape.org/gp/index.html#/main/step1; accessed on 23 January 2022) from 4 comparisons old Atg7^F/F^ vs. young Atg7^F/F^, old atg7ΔKC vs. young atg7ΔKC, young atg7ΔKC vs. young Atg7^F/F^ and old atg7ΔKC vs. old Atg7^F/F^. For selected samples and genes, a heatmap was plotted using the R package pheatmap with the variance-stabilizing transformed count data generated by DESeq2.

### 4.6. Immunofluorescent Staining and Microscopy

Paraffin-embedded tissue sections were cut at a thickness of 5 μm and, after antigen demasking, were incubated overnight at 4 °C with rabbit anti-FABP5 (Cell Signaling Technology, #39926, 1:1200 in PBS with 2% BSA and 10% goat serum for blocking of unspecific binding.

The next day, the sections were washed and incubated with the secondary antibody Alexa Fluor® 546 goat anti-rabbit, A11035 (diluted 1:500 in PBS with 2% BSA), for 30 min at room temperature and counterstained with 2 μg/mL Hoechst-33258 (Molecular Probes, H1398) to visualize the nuclei and mounted in Permafluor (Thermo Scientific, TA-030-FM) for immunofluorescence microscopy. Images were acquired with identical settings using a BX63 upright microscope equipped with a UC90 9-megapixel camera operated via the cellSens software at 40× magnification (all Olympus). Subsequently, the average epidermal FABP5 staining intensity was evaluated in at least 5 fields of view in sections of 3 different mice per genotype and age group. The average fluorescence intensity per field of view was measured within the epidermal area as marked with the polygon selection tool in ImageJ and the statistics within individual and age groups, respectively, were calculated using Microsoft Excel.

## Figures and Tables

**Figure 1 ijms-23-11110-f001:**
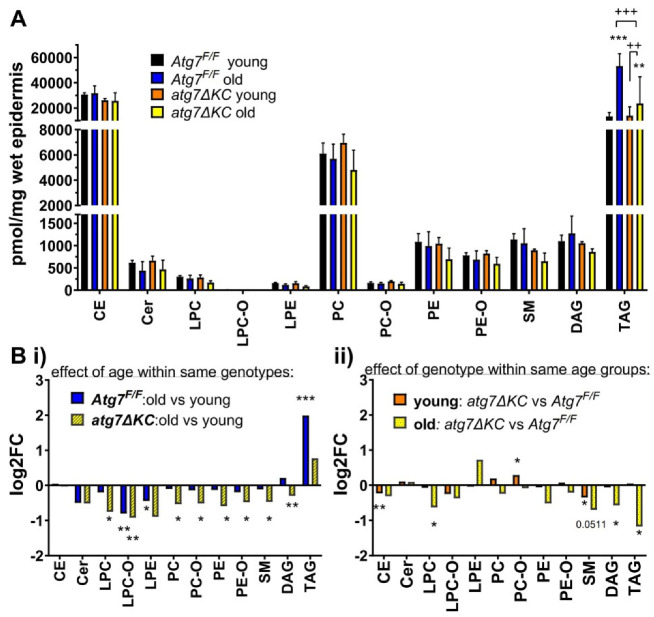
(**A**) Lipid classes identified in murine epidermal extracts. Absolute amounts are expressed as pmol/mg of hydrated epidermal tissue. Two-way ANOVA with Bonferroni’s multiple comparisons test, mean ± standard deviation. * marks a significant difference compared to the control group (Atg7^F/F^ young), + shows a significant difference between two respective groups indicated by a line segment; **/++ *p* < 0.01, ***/+++ *p* < 0.001; *n* = 4 (for Atg7f/f young and old, and atg7ΔKC young) and 5 (for atg7ΔKC old). (**B**) Regulation of lipid classes by age and genotype. Panels show fold change of individual lipid classes (**i**) in aged animals compared to their young counterparts and (**ii**) in animals of different genotypes but of comparable age. Student’s *t*-test, log fold change (mean of replicate values of the given group divided with a mean of replicates of a reference group). * marks a significant difference compared to the reference group; * *p* < 0.05, ** *p* < 0.01; *n* = 4 (for Atg7f/f young and old, and atg7ΔKC young) and 5 (for atg7ΔKC old). CE: cholesteryl ester, Cer: ceramide, LPC: lysophosphatidylcholine, LPC-O: ether-linked LPC, LPE: lysophosphatidylethanolamine, PC: phosphatidylcholine, PC-O: ether-linked PC, PE: phosphatidylethanolamine, PE-O: ethanolamine plasmalogen, SM: sphingomyeline, DAG: diacylglycerol, TAG: triacylglycerol.

**Figure 2 ijms-23-11110-f002:**
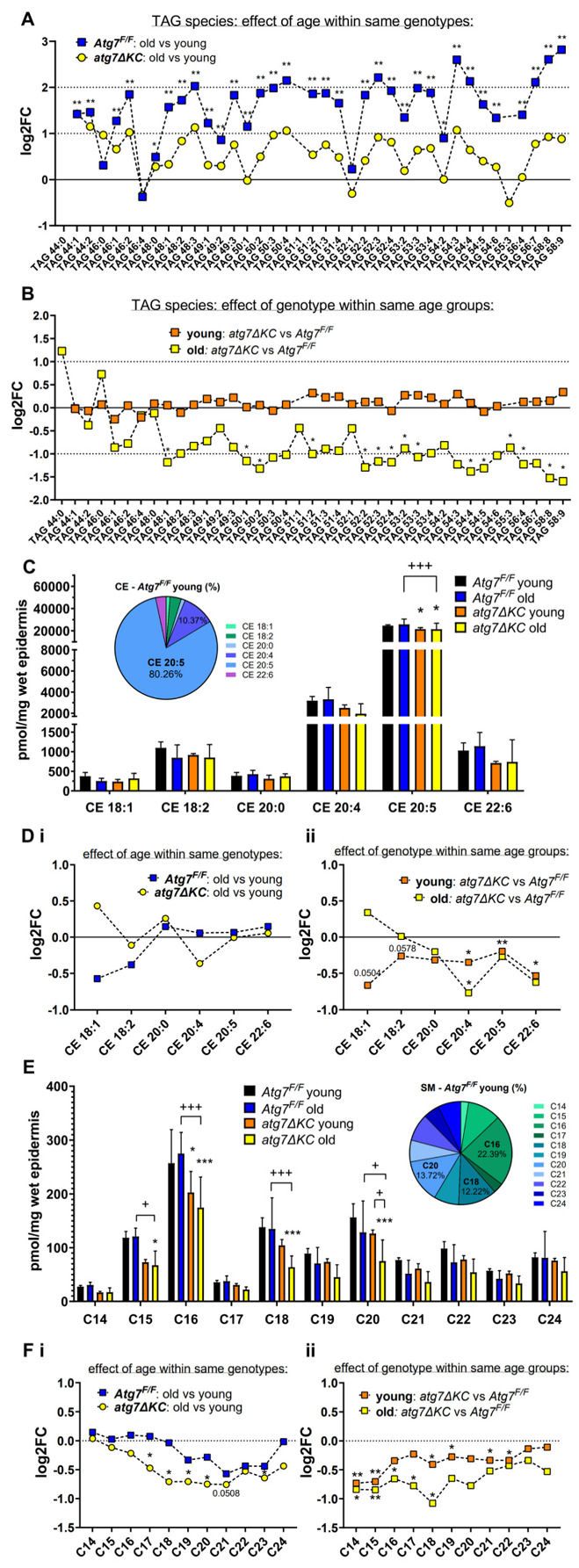
Individual triacylglycerol (TAG) species are mainly upregulated in the tail epidermis of aged mice. B TAG levels remain unchanged in the young knockout epidermis but are decreased in aged knockout, compared to the age-matched control epidermis. Panels show fold change of individual TAG species (**A**) in aged animals compared to their young counterparts and (**B**) in animals of different genotypes but of comparable age. Student’s *t*-test, log fold change (mean of replicate values of the given group divided with a mean of replicates of a reference group). * marks a significant difference compared to the reference group; * *p* < 0,05, ** *p* < 0,01; *n* = 4 (for Atg7f/f young and old, and atg7ΔKC young) and 5 (for atg7ΔKC old). (**C**,**D**) Profile of cholesterol esters (CE) identified in murine epidermal extracts. C Absolute amounts are expressed as pmol/mg of hydrated epidermal tissue. Two-way ANOVA with Bonferroni’s multiple comparisons test, mean ± standard deviation. * marks a significant difference compared to the control group (Atg7^F/F^ young), + shows a significant difference between two respective groups indicated by a line segment; * *p* < 0,05, ** *p* < 0,01, +++ *p* < 0.001; *n* = 4 (for Atg7f/f young and old, and atg7ΔKC young) and 5 (for atg7ΔKC old). The inserted parts of a whole diagram show an individual species proportion within the CE class identified in the young control group. For CE species distribution in other groups studied see Appendix A. (**D**) Panels show fold change of individual CE species (**i**) in aged animals compared to their young counterparts and (**ii**) in animals of different genotype but of comparable age. Student’s *t*-test, log fold change (mean of replicate values of given group divided with a mean of replicates of a reference group). * marks a significant difference compared to the reference group; * *p* < 0.05, ** *p* < 0.01; *n* = 4 (for Atg7f/f young and old, and atg7ΔKC young) and 5 (for atg7ΔKC old). (**E**,**F**) Acyl chain distribution within the sphingomyelin (SM) class was identified. E SMs with identical carbon numbers are combined (assuming that the sphingoid base has 18C, e.g. predicted C16 = SM d34:x); for the complete SM species overview see Appendix A. Absolute amounts are expressed as pmol/mg of hydrated epidermal tissue. Two-way ANOVA with Bonferroni’s multiple comparisons test, mean ± standard deviation. * marks a significant difference compared to the control group (Atg7^F/F^ young), + shows a significant difference between two respective groups indicated by a line segment; */+ *p* < 0,05, ** *p* < 0.01, ***/+++ *p* < 0.001; *n* = 4 (for Atg7f/f young and old, and atg7ΔKC young) and 5 (for atg7ΔKC old). The inserted parts of the whole diagram show the proportion of predicted acyl chain lengths within the SM class in the control group. For SM chain length distribution in other groups studied see Appendix A. (**F**) Panels show fold change of individual SM species with the identical carbon number species (**i**) in aged animals compared to their young counterparts and (**ii**) in animals of different genotype but of comparable age. Student’s *t*-test, log fold change (mean of replicate values of the given group divided with a mean of replicates of a reference group). * marks a significant difference compared to the reference group; * *p* < 0.05, ** *p* < 0.01; *n* = 4 (for Atg7f/f young and old, and atg7ΔKC young) and 5 (for atg7ΔKC old). Note: Data presented in panels (**A**,**B**,**D**,**F**): symbols connected with a dashed line were chosen here only for a better visual depiction of trends or significant patterns and their comparison.

**Figure 3 ijms-23-11110-f003:**
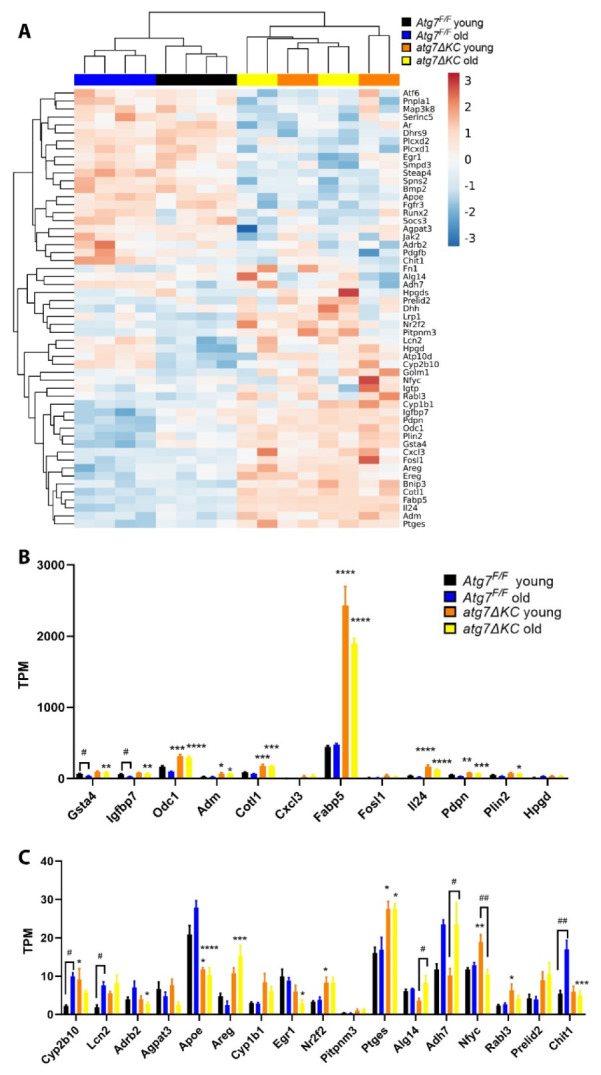
Expression of genes with a function in lipid metabolism-localisation or-signalling in cultured epidermal keratinocytes from young and old Atg7^F/F^ or atg7ΔKC. (**A**) The heatmap of selected 55 lipid-related candidate genes. B and C Bar graph representation of DEG where at least one condition displayed a “Transcripts Per Kilobase Million” (TPM) value of more than 50 (**B**) or between 10 and 50 (**C**). * marks a significant difference between Atg7^F/F^ and atg7ΔKC, # shows a significant difference between young and old groups; */# *p* < 0.05; **/## *p* < 0.01; *** *p* < 0.001; **** *p* < 0.0001; *n* = 4 (for each group).

**Figure 4 ijms-23-11110-f004:**
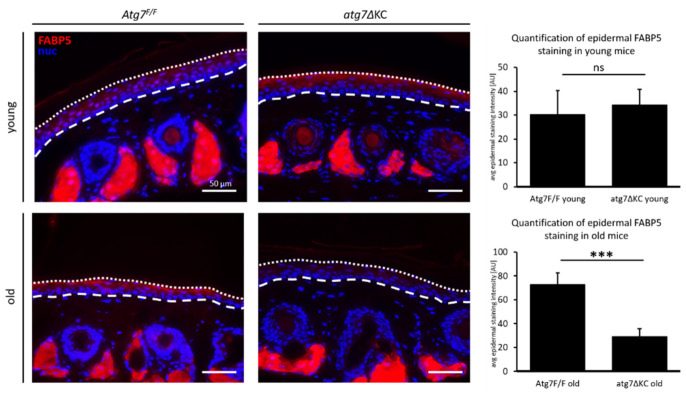
Expression of FABP5 in the tail skin of young and old Atg7F/F or atg7ΔKC mice—epidermal FABP5 is strongly reduced in atg7ΔKC. PFA-fixed, paraffin-embedded tail cross sections were stained with anti-FABP5 antibody and the nuclear dye, Hoechst. Bar graphs show quantification of average epidermal FABP5 staining intensities within the marked areas (dashed lines indicate dermal–epidermal junction and dotted lines, the edge of nucleated areas). Student’s *t*-test, mean + standard deviation. * marks a significant difference between Atg7^F/F^ and atg7ΔKC, ns non-significant; *** *p* < 0,001; *n* = 3 mice per group. Bar: 50 µm.

## Data Availability

Original Lipidomics and RNA seq data are available for reviewers on request from the corresponding author during the reviewing period and RNA seq data will be available from the repository upon acceptance and a quarantine period.

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
