# Peer review of "Consequences of Autophagy Deletion on the Age-Related Changes in the Epidermal Lipidome of Mice"

_ijms, 2022, doi:10.3390/ijms231911110_

Round 1

Reviewer 1 Report

In this study, Yang et al. present a lipidomic analysis of the epidermis from young or aged autophagy-deficient mice (specific to epidermis) compared to control mice. Compared to young control mice, aged control mice have a significantly increased level of triglycerides. This increase in TG storage during aging is largely ablated in autophagy-deficient mice. Other lipid species were also found to be altered in aging autophagy-deficient mice. Upon closer analysis, long-chain polyunsaturated cholesterol esters and long-chain sphingomyelins were reduced in aging autophagy-deficient mice. Transcriptomic analyses from keratinocytes cultured from age-matched mouse epidermis suggest alterations in fatty acid binding proteins, inflammatory markers, and one lipid droplet marker with the highest alteration seen in the fatty acid binding protein Fabp5. However, this finding does not translate in vivo as Fabp5 staining is not increased in atg7DKC mice. On the contrary, it is decreased.

Overall, this study is descriptive. No mechanistic data are presented on how autophagy would influence lipid homeostasis in aged atg7DKC mice. In addition, the transcriptomic data do not support the lipidomic findings or the in situ immunofluorescence staining.

Major points:

·         The transcriptomics samples were collected from cultured cells as opposed to in situ epidermis. The gene expression data seem to oppose the results in vivo. Not only does the Fabp5 staining show the opposite result in vivo compared to the gene expression, but the RNAseq also suggests an increase in Plin2 in aged atg7DKC mouse cells, a lipid droplet coating protein that positively correlates with TG content. It is expected from the lipidomics data that Plin2 expression in aged atg7DKC mice should be lower than aged WT mice.

·         Since RNA transcript levels don’t necessarily correlate with protein levels, protein levels of Fabp5, Plin2, and Apoe should be assessed in keratinocytes and tissue samples of aged atg7DKC mice compared to WT and young controls.

·         Another possible control would be to do RNAseq from in situ epidermis samples where expression levels might correlate better.

·         Given that lipidomics suggests a decrease in TG and CE content in aged atg7DKC mice this should also be confirmed for lipid droplet content by IF analysis for Plin2.

Minor points:

·         Is the capacity for fatty acid uptake altered in aged atg7DKC mouse cells?

·         Figure Legend for Figure 3. Lines 243-245: wrong references to panels (A) (B) should be (B) (C)

·         For the IF staining, what was the antibody incubation buffer, was there a blocking step, and if so what buffer was used?

Author Response

Major points:

  • The transcriptomics samples were collected from cultured cells as opposed to in situ epidermis. The gene expression data seem to oppose the results in vivo. Not only does the Fabp5 staining show the opposite result in vivo compared to the gene expression, but the RNAseq also suggests an increase in Plin2 in aged atg7DKC mouse cells, a lipid droplet coating protein that positively correlates with TG content. It is expected from the lipidomics data that Plin2 expression in aged atg7DKC mice should be lower than aged WT mice.

Rev:        Since RNA transcript levels don’t necessarily correlate with protein levels, protein levels of Fabp5, Plin2, and Apoe should be assessed in keratinocytes and tissue samples of aged atg7DKC mice compared to WT and young controls.

A: We thank the reviewer for this point, and the additional experimentation we made on his/her request indeed brought us forward. A quantification of the FABP5 Protein levels was now performed with imaging software, and this corroborated that in the old Atg7 knockouts the immunoreactivity was significantly lower than in the old autophagy competent tissues. The quantification results are newly displayed in Figure 4. The perceived reduction of the immunoreactivity in old wildtypes compared to young animals was however not confirmed and was likely due to the concentrated staining in the more differentiated strata, as the quantification yielded higher values in the old “WT” epidermis. The respective paragraph (lines 299-305) was now changed:

“The Fabp5 staining intensity was more concentrated in the stratum granulosum of old Atg7F/F animals whereas the epidermis of old atg7ΔKC animals was largely devoid of staining, as shown by quantification with image analysis software (Figure 4). Given the results from the transcriptomic analysis where FABP5 was strongly increased in atg7ΔKC cells (that were however proliferating in culture) this finding was unexpected.”

The corresponding paragraph (lines 409-427) of the discussion was changed accordingly

“Why the FABP5 immuno-reactivity is strongly reduced in the aged tail epidermis sections of atg7ΔKC compared to the controls also remains elusive for now, but suggests that the age dependent effect of autophagy on FABP5 protein depends on differentiation or the epidermal context.  Furthermore, conformational changes upon binding or oxidation (FABP5 is highly prone to oxidation, which can destabilize the protein (45), and we had found more redox stress in atg7ΔKC previously (11) and reduced translation or increased export are hypotheses to be followed in the future.

Whereas this study confirmed that at least in homeostasis epidermal autophagy deficiency does not cause obvious or severe defects in the epidermis in aged mice, it identified that age-related homeostasis of epidermal lipids with known biological activities requires functional autophagy and identified genes dysregulated in autophagy deficient skin that potentially mediate the observed changes in the lipidome or the downstream signaling events.

Our findings provide a rationale for further investigation of the role and the mechanistic base of autophagy-dependent lipid synthesis, mobilization, transport and degradation in regulating inflammation, barrier function and energy metabolism of the epidermis in aging.”

Rev:      Another possible control would be to do RNAseq from in situ epidermis samples where expression levels might correlate better.

A: We agree that RNAseq, or even better - single cell sequencing from in situ epidermis would yield superior data but this is unfortunately beyond the possibilities for this project and not possible within the required response time.

Rev:   Given that lipidomics suggests a decrease in TG and CE content in aged atg7DKC mice this should also be confirmed for lipid droplet content by IF analysis for Plin2.A

A: We agree that localisation of lipids or lipid droplets would be beneficial for such a study. We performed - due to the very short response time very limited and preliminary - experiments to locate with staining neutral lipids in the epidermis in cryosections of old and young mice. We were not able to conduct establishment of Plin2 staining, therefore we performed Bodipy staining on cryosections. These also led to rather confounding results, probably due to contamination with lipids during the cryosectioning process, seen as too much dispersion of lipids/bodipy signal in the sections. That´s why we cannot present any images in the current study but we will follow that up in further studies.

Minor points:

Rev:         Is the capacity for fatty acid uptake altered in aged atg7DKC mouse cells?

A: We thank the reviewer for this excellent suggestion, we will follow that, as many others, in the follow up study, due to time constraints.

Rev:     Figure Legend for Figure 3. Lines 243-245: wrong references to panels (A) (B) should be (B) (C)

A: Thank you, this was corrected. 

Rev:      For the IF staining, what was the antibody incubation buffer, was there a blocking step, and if so what buffer was used?

A: The antibody incubation buffer was PBS/2%BSA and for blocking of unspecific binding of the secondary antibody 10% goat serum were added to the incubation buffer of the primary antibody. This information has now been added to the respective section in Materials and Methods.

Reviewer 2 Report

Major comments

·         The authors haven't justified how targeting only these age groups is the best possible study design to study the questions that they are asking.

· The authors could summarize the results at the end of section 1 more briefly to keep the article engaging from the reader's point of view.

·         In section 2.1, the title doesn't reflect all the results discussed in the section. The authors could rework the title to make it more comprehensive.

·         Also, in section 2.1, the authors need to demonstrate that the deletion of ATG7 occurred specifically in the epidermis and not other tissues of the tail using transcript or protein level analysis.

·         In lines 136-137, the authors should clarify that this reduction was only observed in aged atg7 knockout mice and not young atg7 knockout mice (Figure 1A).

·         Lines 141 -143, the authors should discuss the results of phosphatidylcholine and ether-phosphatidylcholines as they also demonstrate significant differences between knockout young and knockout old mice.

·         Figure 1B, the authors could provide a more nuanced picture of the results by making an age-matched comparison between control and knockout. Comparisons between old knockout mice with control young mice may not be the best.

·         In the legend of figure 1B, the sentence on lines 154-155 could be moved from the figure legend to the results section as it describes the results in the figure.

·         In section 2.2, the authors state that individual TAG species were affected uniformly by age and autophagy deficiency based on the results in Table S1 and Figure S1. However, the results in Figure S1C compare the TAG species abundance between control young mice against old knockout mice. To comment on the effect of autophagy deficiency on TAG species abundance, the authors should compare old knockout mice with control old mice. Also, the authors present a more generalized result for the findings of Table 1 but have only analyzed young control mice vs. old control mice. A better description of the results would be to include the groups compared. Further, the TAG results are also an important part of the study and could be part of the main figures. The authors should consider moving them to the main figures unless these analyses have already been done in some other study. In the latter case, the authors should cite the original study.

·         In Figure 2B the comparisons for fold changes should be done between young and old mice within control and within knockout groups separately. Comparing old knockout mice with control young mice measures changes in two variables (age and gene deficiency. Therefore, this is not a robust way to measure the effect of change in either variable.

·         In Figures 2C and 2D, the interpretation of the decrease in levels of C19 to C23 in aged control mice vs. young control mice may not be correct as the standard deviation in that group is pretty high. Instead, the authors should describe the results of comparisons between young control and knockout mice and knockout control and old mice that provide several interesting and significant insights.

·         In Figures 2C, and 2D, the authors could discuss that Atg7 deletion resulted in an increased reduction in the levels of SM from C17 to C23 significantly.

·         In Figures 2A and 2C, the authors should describe the cholesterol ester (CE) and sphingomyelin (SM) distribution shown inset. If they have data about the distribution of CE in other groups, it would be interesting to check if age or gene deletion affects them.

·         In section 2.3, the results mentioned in lines 224-232 are part of the methodology of how the authors processed the data from the sequencing assay. Therefore, it is best to limit mentioning it as part of materials and methods. The authors could also move them from the legend in Figure 3A. Further, the authors could describe their interpretation of figure 3A in section 2.3.

·         In lines 252 - 263, the authors have described the differentially expressed genes without detailing whether the difference was due to age or gene knockout. Also, the authors have not cited the relevant figure in the text. Further, the authors have not described the results in Figure 3C in this section. Overall, the study would be complete if the authors included their results on the expression levels of the genes that would explain the differences in the TAGs, CEs, and SMs observed in Figures 1 and 2.

·         In section 2.4, the authors need to include a different image or provide quantification to substantiate their claim of a decrease in Fabp5 staining intensity as it is not apparent in the images. Also, the authors have not cited the figure in the text. Further, it would be more helpful to analyze protein expression levels using immunoblotting instead of immunofluorescence.

·         In figure 4, the authors need not include the interpretation of the image about the expression of Fabp5 in the sebaceous gland and epidermal layer.

Minor comments:

·         The authors should provide a citation for the sentences on lines 55-57 and 61-63.

·         On line 86, the authors should mention the details of the cells used in the study while mentioning them for the first time in the manuscript.

·         In figure 4, the authors need to mention the details of the nuclear stain that they used in the legend and the images.

Author Response

Major comments

Rev:      The authors haven't justified how targeting only these age groups is the best possible study design to study the questions that they are asking.

A: Our workgroup had first investigated the epidermal Atg7 knockouts in 10-16 month old animals (Rossiter JDS 2013). Later studies found new and lipid-related phenotypes that were also related to progressed age of the K14Atg7 knockouts (in preputial glands Rossiter Autophagy 2021, in sebaceous glands Rossiter 2018) and also knockouts of Atg7 under the Tyrosinase promoter had a phenotype (also in the brain) that manifested at later age (Sukseree Mol Neurobiol 2018). Thus we wanted to study an epidermal lipid phenotype in younger (below 8 months) and old (older than 16 months) months epidermis, beyond the study group of the initial epidermal investigation.

“We investigated animals beyond the age of 16 months because our recent studies on atg7ΔKC in sebaceous glands and tyrosinase driven atg7 knockouts manifested the most striking and lipid-related phenotypes at advanced age (11, 18).”

Rev:  The authors could summarize the results at the end of section 1 more briefly to keep the article engaging from the reader's point of view.

A: We agree and correspondingly shortened the end of section 1 that now reads

Here we extended these studies to cohorts of young (< 8 mo) and old (>16 mo) mice which did or did not lack epidermal Atg7, and studied the consequences of chronological aging on the level of the lipidome of the epidermis. We investigated animals beyond the age of 16 months because our recent studies on atg7ΔKC in sebaceous glands and tyrosinase driven atg7 knockouts manifested the most striking and lipid-related phenotypes at advanced age (11, 18). We found an age related increase in triglyceride levels in the mouse tail epidermis that was significantly less pronounced in the autophagy deficient old animals. Quantification of the class of cholesterol esters showed a significant decrease in the esterified long chain polyunsaturated fatty acids in the knockouts. The content of sphingomyelins (SM) with long acyl chains (C18-C24) decreased in the aged epidermis, and this effect was strongly enhanced in the autophagy knockouts, which additionally had significantly decreased content in short acyl chain (C13-C17) SM species.

In the transcriptome of corresponding tail epidermal keratinocytes we found the fatty acid transporter Fatty Acid Binding Protein 5 (FABP5) as the gene most highly induced-gene.

Immunohistological analysis of FABP5 expression showed an age-related change in epidermal staining distribution and, in contrast to gene expression in the monolayer cultured keratinocytes, a massive decrease in FABP5 staining in the old Atg7 epidermal knockout animals compared to controls. Together the results demonstrate a role for autophagy in the regulation of triglyceride synthesis and of the fatty acid composition of cholesterol esters and sphingolipids of the epidermis in aging.”

Rev:       In section 2.1, the title doesn't reflect all the results discussed in the section. The authors could rework the title to make it more comprehensive.

A: We agree and changed the title of 2.1 to

“Aged autophagy deficient epidermis accumulates less TAG than aged autophagy competent epidermis and shows dysregulation of other lipid species.”

Rev:       Also, in section 2.1, the authors need to demonstrate that the deletion of ATG7 occurred specifically in the epidermis and not other tissues of the tail using transcript or protein level analysis.

A: The characterisation of the K14:cre Atg7f/f tissues in our mouse strain crossings  is detailed in  H. Rossiter et al. / Journal of Dermatological Science 71 (2013) 67–75.

The general epidermal (including appendages) specificity of the Jackson Labs Tg(KRT14-cre)1Amc line were studied e.g. in Indra et al 2000, Hormone Research, showing epidermal but no dermal or muscle activity of the recombinase (Targeted Somatic Mutagenesis in Mouse Epidermis - Scientific Figure on ResearchGate. Available from: https://www.researchgate.net/figure/Characterization-of-Cre-recombinase-activity-in-K14-Cre-transgenic-lines-A-Structure-of_fig1_11753885)

Rev:    In lines 136-137, the authors should clarify that this reduction was only observed in aged atg7 knockout mice and not young atg7 knockout mice (Figure 1A).

A: Thank you for pinpointing this inaccuracy. We corrected the sentence in the text as follows:

“In the aged atg7ΔKC, the accumulation of TAG in aged controls described above was significantly reduced, however their levels were still significantly upregulated compared to both the young groups (Figure 1A).”

Rev:    Lines 141 -143, the authors should discuss the results of phosphatidylcholine and ether-phosphatidylcholines as they also demonstrate significant differences between knockout young and knockout old mice.

A: Thank you for pinpointing this omittance, we corrected the sentence in the text now:

“Significant decrease only in the old knockouts was observed for lysophosphatidylcholines (LPC), ether-lysophosphatidylcholines (LPC-O), phosphatidylcholines (PC), ether PC (PC-O), phosphatidylethanolamines (PE), ether-PE (PE-O) and lyso-PE (LPE). In the case of LPC-O and LPE a significant but smaller decrease was also observed in the aged controls (Figure 1b).

Rev:    Figure 1B, the authors could provide a more nuanced picture of the results by making an age-matched comparison between control and knockout. Comparisons between old knockout mice with control young mice may not be the best.

A:  Thank you for your suggestion. The information (statistical significance) about comparison between age-matched control and knockout was already included in the original Figure 1B, however we are aware that it was not clearly described and also the graphic presentation was misleading. Therefore, we have changed the panel B of Figure 1. We now present the data in 2 separate graphs: one focusing on differences between aged groups of same genotype, and second, showing differences between genotypes of age-matching groups.

Rev:      In the legend of figure 1B, the sentence on lines 154-155 could be moved from the figure legend to the results section as it describes the results in the figure.

A: Thank you for your suggestion. The sentence from the figure legend was removed as the panel B has been changed. The results in the section 2.1 are newly described as follows:

“Among the lipid classes, we found a striking increase of TAGs in the epidermis of aged control mice as compared to the young controls. In the aged atg7ΔKC, the accumulation of TAG in aged controls described above was significantly reduced, however their levels were still significantly upregulated compared to both the young groups (Figure 1A).”

Rev:        In section 2.2, the authors state that individual TAG species were affected uniformly by age and autophagy deficiency based on the results in Table S1 and Figure S1. However, the results in Figure S1C compare the TAG species abundance between control young mice against old knockout mice. To comment on the effect of autophagy deficiency on TAG species abundance, the authors should compare old knockout mice with control old mice. Also, the authors present a more generalized result for the findings of Table 1 but have only analyzed young control mice vs. old control mice. A better description of the results would be to include the groups compared. Further, the TAG results are also an important part of the study and could be part of the main figures. The authors should consider moving them to the main figures unless these analyses have already been done in some other study. In the latter case, the authors should cite the original study.

A:  Thank you for noticing this misleading expression. We corrected the sentence:

“While the individual TAG species analyzed were affected similarly by aging, meaning that the autophagy deficiency reduced the age-related upregulation of TAG species levels in a uniform way (Figure 2A, B and Supplementary Material, Table S1 and Figure S1),”

We also included new panels into the main Figure2: now showing the fold change of individual TAG species A) in aged animals compared to their young counterparts and B) in animals of different genotype but of comparable age. To the Figure S1 we added panel E and F displaying the TAG total carbon length and total double bounds content in the both aged genotype groups. The Figure legend was also modified, to fit the current Figure. We have expanded the Table S1 with statistical results of comparison of knockout old vs. control old. Other comparisons (knockout young versus control young; knockout old versus knockout young) are only mentioned in the Table legend, as there were no significant differences between them.

Rev:        In Figure 2B the comparisons for fold changes should be done between young and old mice within control and within knockout groups separately. Comparing old knockout mice with control young mice measures changes in two variables (age and gene deficiency. Therefore, this is not a robust way to measure the effect of change in either variable.

A: Thank you for your comment. The results´ presentation style was the same as in the original Figure 1. Now we have changed it so that in the new panels D and F are now the data displayed in 2 separate graphs: one focusing on differences between aged groups of same genotype, and second, showing differences between genotypes of age-matching groups. Here we chose present the log2FC as symbols connected with a dashed lines for a better visual depiction of trends or significant patterns. This information has been also added in the Figure 2 legend as a note (#).

Rev:         In Figures 2C and 2D, the interpretation of the decrease in levels of C19 to C23 in aged control mice vs. young control mice may not be correct as the standard deviation in that group is pretty high. Instead, the authors should describe the results of comparisons between young control and knockout mice and knockout control and old mice that provide several interesting and significant insights.

A: As described above, we have changed the data graphical and statistical presentation for SM acyl chain length (newly Figure2 E, F) and corrected the text in the results section 2.2:

“From the SM species investigated several observations can be made. While the long chain species (C19 – C23) were by trend reduced in the aged Atg7F/F control cohort, the Atg7 deletion resulted in a further (significant) reduction of the same species (C17 – C23) in the aged atg7ΔKC cohort, when compared to the corresponding young counterparts (Figure 2E, F). The atg7ΔKC also showed a significant decrease in the shorter chain SM species, C14 – C18 in old and C14-15, C18-19 and C21-22 in young groups when compared to their age-matching controls, respectively (Figure 2E, F).”

Rev:          In Figures 2C, and 2D, the authors could discuss that Atg7 deletion resulted in an increased reduction in the levels of SM from C17 to C23 significantly.

A: This concern was addressed in the previous answer.

Rev:         In Figures 2A and 2C, the authors should describe the cholesterol ester (CE) and sphingomyelin (SM) distribution shown inset. If they have data about the distribution of CE in other groups, it would be interesting to check if age or gene deletion affects them.

A: Thank you for your comment. We have now included the CE species and SM chain length proportional distribution in Supplementary Info (Figure S2 and S4). For CE species, the only significant difference (t-test) was decrease in CE 22:6 in atg7ΔKC young compared to control young.

The corrected Figure2 legend:

” The inserted parts-of-whole diagram shows an individual species proportion within the CE class identified in the young control group. For CE species distribution in other studied groups see Supplementary Info, Figure S2. … The inserted parts-of-whole diagram shows the proportion of predicted acyl-chain lengths within the SM class in the control group. For SM chain length distribution in other studied groups see Supplementary Info, Figure S4.”

Rev:       In section 2.3, the results mentioned in lines 224-232 are part of the methodology of how the authors processed the data from the sequencing assay. Therefore, it is best to limit mentioning it as part of materials and methods. The authors could also move them from the legend in Figure 3A. Further, the authors could describe their interpretation of figure 3A in section 2.3.

A: Thank you for your suggestions. We have limited the methodology description in section 2.3 and we have moved the description of the method from the legend in Figure 3A accordingly in the manuscript. At the same time, we described more about the interpretation of figure 3A in section 2.3.

Rev:         In lines 252 - 263, the authors have described the differentially expressed genes without detailing whether the difference was due to age or gene knockout. Also, the authors have not cited the relevant figure in the text. Further, the authors have not described the results in Figure 3C in this section.

A: Thank you for your suggestions. We have added the related description detailing whether the difference was due to age or gene knockout and we have added the citation for the relevant figure in the text. Actually, we have described the results in Figure 3C in this section as “Apolipoprotein E (ApoE), a transporter of lipoproteins that is also expressed in basal epidermis (19), on the other hand, was downregulated in autophagy deficient cells” and “and “lipocalin 2 (LCN2), as well as prostaglandin E synthase (PTGES), the products of the latter regulating neutrophil function”.

We have added the citation of Figure 3C at the end of these sentences to make it clearer.

“Among them, PTGES and CXCL3 expression increased in autophagy deficient groups compared with the autophagy competent groups. On the other hand, LCN2 expression increased in old mice compared with young ones. IL24 and GSTA4 expression were affected by both age and autophagy competence and showed increases in autophagy deficient groups and decreases in old groups (Figure 3B, C).”

Rev:         Overall, the study would be complete if the authors included their results on the expression levels of the genes that would explain the differences in the TAGs, CEs, and SMs observed in Figures 1 and 2.

A: All DEGs with lipid-related function are shown; therefore differences between TAGs, CEs, and SMs that are not affected by those genes may derive from gene expression independent/indirect effects of autophagy in the differentiating epidermis.

Rev:       In section 2.4, the authors need to include a different image or provide quantification to substantiate their claim of a decrease in Fabp5 staining intensity as it is not apparent in the images. Also, the authors have not cited the figure in the text. Further, it would be more helpful to analyze protein expression levels using immunoblotting instead of immunofluorescence.

A: We thank the reviewer for the important concern, and we performed an Image Analysis Software Quantification now of at least 5 FOV form at least 3 animals of each group, and the results prompted us to rephrase the findings (this is also an answer to Reviewer 1),

“The Fabp5 staining intensity was more concentrated in the stratum granulosum of old Atg7F/F animals whereas the epidermis of old atg7ΔKC animals was largely devoid of staining, as shown by quantification with image analysis software (Figure 4). Given the results from the transcriptomic analysis where FABP5 was strongly increased in atg7ΔKC cells (that were however proliferating in culture) this finding was unexpected.”

The corresponding paragraph (lines 409-427) of the discussion was changed accordingly

“Why the FABP5 immuno-reactivity is strongly reduced in the aged tail epidermis sections of atg7ΔKC compared to the controls also remains elusive for now, but suggests that the age dependent effect of autophagy on FABP5 protein depends on differentiation or the epidermal context.  Furthermore, conformational changes upon binding or oxidation (FABP5 is highly prone to oxidation, which can destabilize the protein (45), and we had found more redox stress in atg7ΔKC previously (11) and reduced translation or increased export are hypotheses to be followed in the future.

Whereas this study confirmed that at least in homeostasis epidermal autophagy deficiency does not cause obvious or severe defects in the epidermis in aged mice, it identified that age-related homeostasis of epidermal lipids with known biological activities requires functional autophagy and identified genes dysregulated in autophagy deficient skin that potentially mediate the observed changes in the lipidome or the downstream signaling events.

Our findings provide a rationale for further investigation of the role and the mechanistic base of autophagy-dependent lipid synthesis, mobilization, transport and degradation in regulating inflammation, barrier function and energy metabolism of the epidermis in aging.”

We agree that an alternative method of protein quantification would be ideal, but this was not possible in the given response time.

Rev:       In figure 4, the authors need not include the interpretation of the image about the expression of Fabp5 in the sebaceous gland and epidermal layer.

A: We agree and we deleted the corresponding sentence from the figure legend.

Minor comments:

Rev:        The authors should provide a citation for the sentences on lines 55-57 and 61-63.

A: Thank you, the references were included.

Rev:        On line 86, the authors should mention the details of the cells used in the study while mentioning them for the first time in the manuscript.

A: As indicated in the Material and Methods section, these are primary keratinocytes isolated from tail skin of respective mouse.

Rev:        In figure 4, the authors need to mention the details of the nuclear stain that they used in the legend and the images.

A: The details of Hoechst nuclear dye are specified in the Material and Method section.

Reviewer 3 Report

The submitted manuscript is well-designed and performed, and the results are also reasonably interpreted.

- Considering the potential roles of triacylglycerol in epidermis as an energy storage and supplier for fatty acids moieties into other lipids, it would be interesting to investigate the changes of ceramides, especially in stratum corneum. 

- Did authors measure the functional parameters of skin, such as skin hydration and epidermal permeability barrier function? In previous studies using asebia mouse model [J Invest Dermatol (2003) 120(5); 728], decreased skin hydration was observed, and similar functional changes might be observed in currently used model. It also might explain the potential relationship between the insufficient autophagy activity and skin dysfunctions in aged skin. 

Author Response

Rev:           Considering the potential roles of triacylglycerol in epidermis as an energy storage and supplier for fatty acids moieties into other lipids, it would be interesting to investigate the changes of ceramides, especially in stratum corneum. 

A: Thank you for your great suggestion. We have investigated some of ceramide species (limited by the used analytical method) from the whole epidermal extract. The identified species, their total amount and their proportional distribution can be now found in Supplementary Info, Figure S5. We also newly comment on the Cer species in Results section 2.2:

“As SMs are one of the precursors of ceramides (Cer) (19), and although total amount of Cer was affected neither by age nor by autophagy knockout (Figure 1A), Cer species with rather short acyl chain (C18 and C20) were increased and Cer with long acyl chain (C24) decreased in atg7ΔKC young epidermis (Supplementary Material, Figure S5), which could partially explain the observed changes in SM levels.”

We also agree that focusing on the stratum corneum and its barrier properties is worth following up in next studies.

Rev:            Did authors measure the functional parameters of skin, such as skin hydration and epidermal permeability barrier function? In previous studies using asebia mouse model [J Invest Dermatol (2003) 120(5); 728], decreased skin hydration was observed, and similar functional changes might be observed in currently used model. It also might explain the potential relationship between the insufficient autophagy activity and skin dysfunctions in aged skin. 

A: Thank you for your comment. The permeability barrier of Atg7 knockout skin of newborn and adult mice was investigated in our first paper: Rossiter JDS 2013. In that study no differences in permeability properties were described. We did not measure functional parameters of skin in the current study, as we worked only with the tail skin. However, it could be a part of a follow up study focusing on skin barrier in these advanced aged mice.

Round 2

Reviewer 1 Report

I appreciate that the authors took care to address my concerns.

Most questions have been answered to my satisfaction and I note the time constraints that are preventing further experimentation.

There are a few minor notes for Figure 4.

1. Please add the objective magnification used for acquiring the images to the imaging methods. 

2. The units for fluorescence quantification are confusing. Does the measurement represent the integrated fluorescence intensity in the measurement field or the average intensity normalized by area of interest?

3. Please add white line markings that outline the measurement area (epidermis) in the images.

4. Were your cryosections postfixed with PFA after thawing? A brief wash with 50:50 MeOh:Acetone could also remove TG leaking from unfixed broken Lipid droplets.

Author Response

Comments and Suggestions for Authors

I appreciate that the authors took care to address my concerns.

Most questions have been answered to my satisfaction and I note the time constraints that are preventing further experimentation.

There are a few minor notes for Figure 4.

Rev:        1. Please add the objective magnification used for acquiring the images to the imaging methods. 

A: Thank you for pointing out this lack of detail. The use of a 40x objective was added in the imaging methods.

Rev:        2. The units for fluorescence quantification are confusing. Does the measurement represent the integrated fluorescence intensity in the measurement field or the average intensity normalized by area of interest?

A: Thank you for flagging this point of confusion. The units have been changed to arbitrary units. The values shown represent the mean for individual groups calculated from the average fluorescence intensity within the measured area (the epidermis) per field of view.

Rev:        3. Please add white line markings that outline the measurement area (epidermis) in the images

A: Thank you. A dotted white line has been added at the outer edge of the nucleated part of the epidermis while a dashed line marks the dermal/epidermal junction. Both lines together define the area in which the measurements were taken.

Rev:        4. Were your cryosections postfixed with PFA after thawing? A brief wash with 50:50 MeOh:Acetone could also remove TG leaking from unfixed broken Lipid droplets.

A: Thank you for your suggestion. All stainings shown in this manuscript were performed on formalin fixed, paraffin embedded sections.

Reviewer 2 Report

The authors Yang & Kremslehner et al. have submitted a revision to the manuscript titled "Consequences of autophagy deletion on the age-2 related changes in the epidermal lipidome of mice". The authors did a good job of improving the manuscript and responding to all the queries satisfactorily. Following are the comments on the revised manuscript:

Major comments:

·         In figure 1B, 2D, and 2F, the current graph type is more beneficial for analyzing the effect of genotype on changes in levels of lipids between young and old mice (Figure 1Bi, 2Di, 2Fi) and the effect of age on changes in levels of lipids between control and KO mice (Figure 1Bii, 2Dii). Figure 1A sufficiently demonstrates the differences that the authors described in section 2.1. Therefore, they should clarify the analysis that they are trying to present. Also, authors should make comparisons for statistical analysis between the groups included in those graphs to depict the results accurately.

·         Each result sub-section could end with a summary of the results described above, followed by a conclusion that leads the reader to the subsequent results.

·         In the legend of Figure 1, the number of mice used could be revised as n = 4 (for young Atg7f/f young and old, and  atg7ΔKC young) and 5 (for atg7ΔKC old) if this is what the authors meant. If not, please revise to clarify the number of mice for each group. Also, please make these revisions across the manuscript.

·         The title of section 2.2 should be revised to include details of updated Figure 2. Also, in the main text, it would be more beneficial to describe each sub-figure of Figure 2 separately as the parameters differ. The authors could follow up such a description with a summary of a couple of sub-figures depicting results of the same lipid species.

·         In line 168, the statement contradicts the results in supplementary table 1 (t-test column between old control and old knockout mice), and supplementary Figure 1E shows a significant difference only in some TAG species. Therefore, the authors should revise their results.

·         In lines 170 – 172, the authors need to provide a more nuanced description of Figures 2C and 2D as the results are not the same across the different CE and the tested groups. Also, the authors have erroneously mentioned that CE 20:5 has a decrease by trend instead of a significant reduction. Instead, figure 2C shows that CE 20:4 has decreased by trend while CE 20:5 has a considerable decline. Further, the authors should discuss here or in discussion how the decrease in levels of these species of CE affects aging. This discussion would help to provide more significance in testing the levels of individual species of CE vs. checking only the total levels of CE.

·         In the description of Figures 2E, the authors mention C19 to C23 as long chain species but later include C17 and C18 in these species. It would be helpful if the authors kept their classification uniform. Also, they mention that SM species C17/C19 to C23 all decrease by a trend in old control mice and significantly in old knockout mice compared to young control mice. However, this is only true for C18 and C20. All the other species have a high standard deviation that negates the decrease by trend. The authors should revise this observation.

·         In lines 179-182, the authors mention a significant decrease in the shorter chain SM species C14-C18. However, figure 2E shows that the significant difference is not present in groups C14 and C17. Similarly, only C16 shows a considerable difference between the young KO and young control.

·         In lines 185 - 187, the groups of ceramides do not match the groups mentioned in supplementary figure S5. The authors should revise the text or the graph.

·         In section 2.3, the authors mention a significant change in the expression of CXCL3. However, Figure 3B shows no such differences. The authors should re-check the levels and revise the statements accordingly.

·         In Figure 4 legend, the authors mention n=4/5 mice per group. However, in the materials and methods section, the authors say n = 3 mice per group. Please incorporate the correct version at both locations.

·         The discussion or end of section 2.1 should justify the differences in the changes in the levels of TAGs as compared to CE and SM over age and genotype.

Minor comments:

·         In line 42, the wording for ATG genes should follow the terminology suggested in PMID:22889836.

·         In lines 109-110, the group being discussed is not clear. The authors need to rework this part.

·         In lines 123-125, Please revise the wording of the controls. The current version mentions that the Cre-LoxP system deletes Atg7 in both KO and control.

·         In lines 134-136, The current version of the sentence jumbles the inference of the result. Please reword it.

·         In line 143, authors could rename Ether-lysophosphatidylcholines (LPC-O) like ether-PC (PC-O).

·         Each sub-figure title should have the same formatting across all figure legends.

Author Response

Major comments:

Rev:         In figure 1B, 2D, and 2F, the current graph type is more beneficial for analyzing the effect of genotype on changes in levels of lipids between young and old mice (Figure 1Bi, 2Di, 2Fi) and the effect of age on changes in levels of lipids between control and KO mice (Figure 1Bii, 2Dii). Figure 1A sufficiently demonstrates the differences that the authors described in section 2.1. Therefore, they should clarify the analysis that they are trying to present. Also, authors should make comparisons for statistical analysis between the groups included in those graphs to depict the results accurately.

 A: Thank you for your comment. We now state clearer in the main text what different panels of Figure 1 represent.

“Next, we analyzed in detail the changes within the individual lipid classes, focusing on differences between aged animals and their young counterparts (Figure 1B i) and between animals of different genotype but of comparable age (Figure 1B ii). We found that there was a decrease in cholesterol esters (CE, significant only for young knockouts) and in sphingomyelins (SM) for the knockouts of both age groups when compared to the age-matched controls (Figure 1B ii).”

Rev:         Each result sub-section could end with a summary of the results described above, followed by a conclusion that leads the reader to the subsequent results.

 A: Thank you, we agree with the comment. We have modified the paragraph 2.1 and paragraph 2.2. Please see corresponding parts in the revised manuscript.

Rev:        In the legend of Figure 1, the number of mice used could be revised as n = 4 (for young Atg7f/f young and old, and  atg7ΔKC young) and 5 (for atg7ΔKC old) if this is what the authors meant. If not, please revise to clarify the number of mice for each group. Also, please make these revisions across the manuscript.

 A: Thank you for your suggestion. We revised the manuscript and specified number of samples used for analyses. Note that Figure 1, 2 and 4 show results from epidermis, Figure 3 are transcriptome data from cultured cells, therefore there are differences in the sample numbers.

Rev:         The title of section 2.2 should be revised to include details of updated Figure 2. Also, in the main text, it would be more beneficial to describe each sub-figure of Figure 2 separately as the parameters differ. The authors could follow up such a description with a summary of a couple of sub-figures depicting results of the same lipid species.

 A: Thank you, we modified the title as follows:

“Autophagy deficiency decreases TAG unselectively but leads to selective reduction of epidermal CE esterified with long chain PUFA and to reduced abundance of SM species”.

We also newly described each panel of Figure 2 separately and included a short discussion into section 2.2. For the all changes done please see the revised manuscript, Results, section 2.2

Rev:         In line 168, the statement contradicts the results in supplementary table 1 (t-test column between old control and old knockout mice), and supplementary Figure 1E shows a significant difference only in some TAG species. Therefore, the authors should revise their results.

 A: Thank you for your comment. Figures 2A,B shows all individual identified species, on contrary the supplementary Figure S1E showed only species with the same total chain length (e.g. to the total length of 50C contributes four different TAG species). The original aim of showing Figure S1 was just to visualize that the patterns of changes are very similar. However, we have newly deleted Supplementary Figure S1, as the more detailed results are included in the main Figure 2 A and B and the double presentation could be confusing also for readers.  

Figure 2B (yellow symbols) shows the same data as are presented in the Supplementary Table I, t-test comparison of aged knockout vs aged control. We also specify the reference to Supplementary Table I.

“A closer evaluation within lipid classes showed that autophagy deficiency reduced the age-related upregulation of most of the TAG species in a similar way Figure 2A, B (for mean values ± SD see Supplementary Material, Table I). This could suggest that the autophagy deficiency either leads to increased TAG lipolysis or to their decreased synthesis and therefore decreased accumulation.”

Rev:         In lines 170 – 172, the authors need to provide a more nuanced description of Figures 2C and 2D as the results are not the same across the different CE and the tested groups. Also, the authors have erroneously mentioned that CE 20:5 has a decrease by trend instead of a significant reduction. Instead, figure 2C shows that CE 20:4 has decreased by trend while CE 20:5 has a considerable decline. Further, the authors should discuss here or in discussion how the decrease in levels of these species of CE affects aging. This discussion would help to provide more significance in testing the levels of individual species of CE vs. checking only the total levels of CE.

 A: … this inconsistency is caused by different statistical methods used in panels E and F. The text in the Results section describes Figure2D, where t-test was used for comparing always the two stated groups. We have deleted the annotation to Figure 2 panel C from the text, so only Fig 2D is connected to the regulated species description in Results section.

“… that among the CE species investigated there was a significant decrease in the polyunsaturated CE 20:4, CE 20:5 and CE 22:6 in the autophagy deficient epidermal samples (Figure 2 D ii; CE 20:5 and CE 22:6 were reduced only by trend in the old group).”

Also from Figure 2D i) and ii) it is apparent that the changes are age-independent, however similar decrease in PUFA-esterified cholesterol was observed for both autophagy deficient groups. Changes in PUFAs related to autophagy deficiency we discuss in paragraph 3.3.

Rev:         In the description of Figures 2E, the authors mention C19 to C23 as long chain species but later include C17 and C18 in these species. It would be helpful if the authors kept their classification uniform. Also, they mention that SM species C17/C19 to C23 all decrease by a trend in old control mice and significantly in old knockout mice compared to young control mice. However, this is only true for C18 and C20. All the other species have a high standard deviation that negates the decrease by trend. The authors should revise this observation.

A: Thank you. We have reformulated the paragraph, labeling as short acyl chain SM only the species with C14 and C15 and not making any categorization about the other chain lengths. We also revised the corresponding parts in the whole manuscript (e.g. at the end of introduction). For the complete corrected paragraph please see the answer to the next point.

Rev:         In lines 179-182, the authors mention a significant decrease in the shorter chain SM species C14-C18. However, figure 2E shows that the significant difference is not present in groups C14 and C17. Similarly, only C16 shows a considerable difference between the young KO and young control.

A: Thank you for your comment. This inconsistency was caused by different statistical methods used in panels E and F. The text in the Results section of old manuscript version described Figure2F, where t-test was used for comparing always the two stated groups. To avoid misleading interpretation, we have specified annotations to Figure 2 panel E and Figure 2 panel F i) or ii). We also comment on their results depending on the statistical method used.

“From the SM species investigated several observations can be made. Firstly, there is an apparent trend towards reduction in their absolute abundance in both knockout groups for all SMs, with a stronger effect in the aged cohorts, and for acyl chain lengths C14 - C20 (significant for C15, 16, 18 and 20; Figure 2E, analysis of variance). These changes are also visible in Figure 2F (effect of either age or genotype): the species with acyl chain length of C19 – C23 were by trend reduced in the aged Atg7F/F control cohort, the Atg7 deletion resulted in a further (significant) reduction of the same species and of species with C17 and C18 in the aged atg7ΔKC cohort, when compared to the corresponding young counterparts (Figure 2F i). The atg7ΔKC also showed a significant age-independent decrease in the short chain SM species, C14 and C15 (Figure 2F ii). Furthermore, SM C16 - C18 in aged and C18, 19, 21, 22 in young knockouts were significantly decreased when compared to their age-matching controls (Figure 2F ii).”

Rev:         In lines 185 - 187, the groups of ceramides do not match the groups mentioned in supplementary figure S5. The authors should revise the text or the graph.

A: Thank you. We added a new panel to Figure S5 which corresponds with the acyl chain length described in the main text. We now also specify the reference to the Figure 5S B.

Rev:         In section 2.3, the authors mention a significant change in the expression of CXCL3. However, Figure 3B shows no such differences. The authors should re-check the levels and revise the statements accordingly.

A: Thank you for addressing this issue. We have revised the statements in the manuscript accordingly as follows:

“Among them, PTGES and CXCL3 (only by trend) expression increased in autophagy deficient groups compared with the autophagy competent groups.”

Rev:         In Figure 4 legend, the authors mention n=4/5 mice per group. However, in the materials and methods section, the authors say n = 3 mice per group. Please incorporate the correct version at both locations.

A: Thank you. The inconsistencies regarding the numbers of mice per group that were used for image analysis has been corrected. Both the figure legend and the method section now mention 3 mice per group.

Rev:         The discussion or end of section 2.1 should justify the differences in the changes in the levels of TAGs as compared to CE and SM over age and genotype.

A: Thank you for your suggestion. We have now added summary of Figure 1 at the end of Results section 2.1. However, we want to avoid speculating on the mechanistic base of the lipid classes being differently affected, because that would require the more holistic analysis of metabolomics etc. and it is beyond the scope of this study.

“In summary, we observed a general decrease in phospholipid classes in the old knock-out but not old control mice compared to their genotype matched controls. On the other hand, in age-matched groups, Atg7 deficiency reduced levels of glycerolipids. As not only the total amount but also the lipid acyl chain length and its saturation influence the physico-chemical behavior and biological function of lipids, we further analysed the species within the regulated lipid classes.”

Minor comments:

Rev:         In line 42, the wording for ATG genes should follow the terminology suggested in PMID:22889836.

 A: Thank you for pointing out this incorrect use of the abbreviation. We corrected it in the text.

 “…depends on autophagy related (Atg) genes which code for a machinery that…”

Rev:         In lines 109-110, the group being discussed is not clear. The authors need to rework this part.

A: Thank you. We added the specification of genotype group.

“In the transcriptome of corresponding tail epidermal keratinocytes we found the fatty acid transporter Fatty Acid Binding Protein 5 (FABP5) as the gene most highly induced in Atg7 depleted tissue.”

Rev:         In lines 123-125, Please revise the wording of the controls. The current version mentions that the Cre-LoxP system deletes Atg7 in both KO and control.

 A: Thank you. We agree that the sentence could be misleading. We rephrased it as follows:

“In this study we investigated tail skin of mice in which the Krt14 (Keratin 14) promoter driven Cre-LoxP system deletes a floxed exon of the essential autophagy gene, Atg7 (atg7ΔKC, knockout) in comparison to tail skin of control mice (Atg7F/F, controls).”

Rev:         In lines 134-136, The current version of the sentence jumbles the inference of the result. Please reword it.

 A: Thank you. We rephrased the sentence as follows:

“This age-related accumulation of TAG was significantly reduced in the aged atg7ΔKC, however their levels were still significantly upregulated compared to the both young groups (Figure 1A).”

Rev:         In line 143, authors could rename Ether-lysophosphatidylcholines (LPC-O) like ether-PC (PC-O).

 A: Thank you for your comment. We unified the abbreviating after the first appearance. However, LPC-O and PC-O are two different subclasses which were analysed separately, therefore we cannot rename LPC-O as suggested.

Rev:         Each sub-figure title should have the same formatting across all figure legends.

A: Thank you for noticing this discrepancy. We revised the formatting and unified it.

Round 3

Reviewer 1 Report

I am satisfied with the changes and added details in response to my comments. 

Just a technical note: BODIPY staining is not compatible with Paraffin embedded tissues the paraffin removal with Xylene and EtOH depletes triglycerides and can alter lipid droplet morphology. If you decide to to look at lipid droplets in situ in this context in the future I would suggest embedding the tissues in OCT and freezing them.

Reviewer 2 Report

The authors did a great job in revising the manuscript.